# Spectro 1—A Potential Spectrophotometer for Measuring Color and Myoglobin Forms in Beef

**DOI:** 10.3390/foods11142091

**Published:** 2022-07-14

**Authors:** Chandler D. Stafford, Mackenzie J. Taylor, David S. Dang, Eric M. England, Daren P. Cornforth, Xin Dai, Sulaiman K. Matarneh

**Affiliations:** 1Department of Nutrition, Dietetics and Food Sciences, Utah State University, Logan, UT 84322, USA; chandler.stafford@usu.edu (C.D.S.); kenzie.taylor@usu.edu (M.J.T.); ds.dang24@gmail.com (D.S.D.); emengland@gmail.com (E.M.E.); daren.cornforth@usu.edu (D.P.C.); 2Utah Agricultural Experiment Station, Utah State University, Logan, UT 84322, USA; xin.dai@usu.edu

**Keywords:** Color Muse Spectro 1, HunterLab MiniScan XE Plus, beef color, myoglobin redox forms, precision

## Abstract

The objective of this study was to compare the Color Muse Spectro 1 sensor to the HunterLab MiniScan XE Plus spectrophotometer for evaluating beef color. Color coordinates (lightness (*L**), redness (*a**), yellowness (*b**), chroma (*C**), and hue (*h**)), myoglobin redox forms (metmyoglobin (MMb), deoxymyoglobin (DMb), and oxymyoglobin (OMb)), and metmyoglobin reducing ability (MRA) were measured on beef steaks over a 5-days storage period. The results indicated that *L**, *b**, *C**, MMb%, OMb%, and MRA% values obtained with Spectro 1 were comparable to those of MiniScan. However, Spectro 1 values for *a** were overestimated compared to MiniScan (*p* < 0.05), whereas those for *h** and DMb% were underestimated (*p* < 0.05). Regardless, Spectro 1 had the capability to detect changes in color coordinates, myoglobin forms, and MRA throughout the storage period. Bland–Altman plots demonstrated that *L**, *b**, and *C** are interchangeable between the two instruments, but it was not the case for *a**, *h**, myoglobin forms, and MRA. Color coordinates measured by Spectro 1 exhibited excellent stability over time, evidenced by the low total color difference (Δ*E***_ab_*) values. Collectively, these findings indicate that Spectro 1 is a potential alternative spectrophotometer for studying meat color and myoglobin redox forms.

## 1. Introduction

The visual appeal of meat is influenced by its color, shape, size, and packaging aesthetics. With regards to fresh beef products, surface color is the most critical. Color is often perceived by consumers as an indicator of beef freshness and wholesomeness at the point of sale [1]. However, beef products are prone to color deterioration during retail display, which can negatively influence consumers intent to purchase them. Greene et al. [2] indicated that most consumers reject retail beef when surface browning exceeds 40%, resulting in an annual revenue loss of approximately one billion USD to the US beef industry [3]. This economic loss has encouraged meat researchers to explore different methodologies for evaluating color stability and mitigating discoloration in beef. While there are many factors that can influence beef color stability, it is well known that discoloration is associated with changes in myoglobin chemistry during storage and display [4].

Myoglobin is the primary pigment driving fresh meat color. It is composed of a single polypeptide chain bound to a heme prosthetic group. The heme portion of myoglobin contains a single iron atom capable of binding ligands. The color of meat is dictated by the nature of the ligand and the valence state of the iron in myoglobin [5]. Oxymyoglobin (OMb) possesses an oxygen bound iron and contributes to the desirable cherry-red color of fresh beef products, whereas myoglobin that lacks oxygen, deoxymyoglobin (DMb), is associated with the purplish-red color typically seen in vacuum-packaged beef [6]. Unfavorable beef discoloration occurs when the iron is oxidized from Fe^2+^ to Fe^3+^, creating a myoglobin derivative known as metmyoglobin (MMb). Oxidized myoglobin imparts a brown color that is visually off-putting to consumers [7]. To that end, proportions of myoglobin forms and surface color are typically measured to assess fresh beef color stability.

Spectroscopy has been widely utilized to evaluate meat color and proportions of myoglobin redox forms [8]. Specifically, hand-held spectrophotometers are commonly used in meat research due to their ease of use and portability [9,10]. However, reputable spectrophotometers frequently cost upwards of several thousand USD, which may not be an option for some researchers. Thus, alternative devices that are cost-effective have recently gained attention among meat scientists [11,12,13]. The Color Muse Spectro 1 (Variable, Inc., Chattanooga, TN, USA; Figure 1) is a pocket-size, hand-held device capable of measuring several different color coordinates such as Hunter *L* (lightness), *a* (redness), and *b* (yellowness) and Commission Internationale de l’Eclairage’s (CIE) *L** (lightness), *a** (redness), and *b** (yellowness) while performing spectral analysis between 400–700 nm. The device is easily operated with the Spectro application (Variable, Inc., Chattanooga, TN, USA) on the user’s smartphone. Collected data are automatically uploaded and stored into a cloud storage service provided by the manufacturer, which improves data accessibility and sharing. More importantly, this device is significantly lower in cost (~USD 300) relative to other devices with similar capabilities, such as the Minolta (Osaka, Japan) and HunterLab (Reston, VA, USA) instruments.

The original intended use of Spectro 1 was for paint, textiles, flooring, and home decor product color assessment and matching. However, this sensor has been recently tested for color measurements in pork, showing promising reliability and reproducibility [14]. To the best of our knowledge, however, no other studies have utilized Spectro 1 for color evaluation in beef, nor has it ever been used to analyze myoglobin redox forms. Therefore, the objective of this study was to investigate the efficacy of the Color Muse Spectro 1 in measuring fresh beef surface color and myoglobin redox forms in comparison to the HunterLab MiniScan XE Plus spectrophotometer (HunterLab Associates, Reston, VA, USA).

## 2. Materials and Methods

### 2.1. Experimental Design

Twenty-four beef steaks measuring 2.54 cm in thickness were purchased from a local grocery store and randomized into two groups (*n* = 12 per group). Group 1 was allocated for spectrophotometric and colorimetric analyses, while group 2 was designated for metmyoglobin reducing ability (MRA) measurements. Each steak was placed on a Styrofoam tray overwrapped with oxygen-permeable polyvinyl chloride film and kept in a refrigerator at 4 °C for 5 days. The aforementioned measurements were evaluated on each steak at 0, 3, and 5 days of storage. At each time point, a sample (2.5 × 2.5 × 2 cm) was removed from each of the group 2 steaks for MRA evaluation.

### 2.2. Colorimetric and Spectrophotometric Measurements

Colorimetric and spectrophotometric measurements were collected using two different instruments: Color Muse Spectro 1 (illuminant A, 10° observer angle, and 8 mm aperture diameter) and HunterLab MiniScan XE Plus (illuminant A, 10° observer angle, and 8 mm aperture diameter). Prior to measurements, the instruments were calibrated according to the respective manufacturer’s instructions. Steaks were removed from their packages and allowed to bloom for 20 min at room temperature. Each steak was placed on a white plastic board and seven replicate scans were collected at random locations. CIE color coordinates (*L**, *a**, and *b**) were recorded and utilized to calculate chroma (*C**; $\sqrt{a^{*2}+b^{*2}}$) and hue angle (*h**; tan^−1^ [*b**/*a**]). For Spectro 1, spectrophotometric measurements (400 to 700 nm at 10 nm intervals) were collected simultaneously with the colorimetric measurements, whereas separate measurements had to be performed to collect spectrophotometric measurements with MiniScan. Reflectance (R) values at 473, 525, 572, and 700 nm were used to calculate reflex attenuance (A) = log(1/R). Percentages of myoglobin forms (MMb, DMb, and OMb) were then calculated using the following equations: %MMb = {1.395 − [(A572 − A700) ÷ (A525 − A700)]} × 100; %DMb = {2.375 × [1 − (A473 − A700) ÷ (A525 − A700)]} × 100; %OMb = {100 − (%MMb + %DMb)} [9,15].

### 2.3. Metmyoglobin Reducing Ability

Metmyoglobin reducing ability was determined following the procedure of Sammel et al. [16]. In brief, the meat portion designated for MRA was oxidized in 0.3% (*w*/*v*) sodium nitrite solution for 20 min at room temperature. Subsequently, samples were removed from the solution, blotted dry, and vacuum packaged. Immediately after packaging, three spectrophotometric scans (400 to 700 nm at 10 nm intervals) were collected from each sample by both instruments over the vacuum package. These values were recorded and used to calculate the initial MMb% using the aforementioned equations. Samples were then placed in an incubator at 30 °C for 2 h and rescanned to calculate the final MMb%. The MRA% was calculated as %MMb reduced: [(initial MMb% − final MMb%) ÷ initial MMb%] × 100.

### 2.4. Instrument Stability

To compare the stability of the instruments over time, *L**, *a**, and *b** were collected with both instruments on standard white, blue, and green calibration tiles provided by the Spectro 1 manufacturer (Variable, Inc., Chattanooga, TN, USA). Immediately after calibration, each tile was scanned seven times at 5 min intervals for a period of 60 min. Total color difference (Δ*E***_ab_*) was then calculated as the difference in color coordinates at each time point from the values at time zero using the following equation:$$\Delta {E^{*}}_{ab}=\sqrt{{\left(L_{2}^{*}-L_{1}^{*}\right)}^{2}+(a_{2}^{*}-a_{1}^{*}{)}^{2}+(b_{2}^{*}-b_{1}^{*}{)}^{2}}$$

### 2.5. Statistical Analysis

All statistical procedures were performed using JMP (version 16.1; SAS Institute, Inc., Cary, NC, USA). Color coordinates, proportions of myoglobin forms, and MRA% data were analyzed using a mixed model for repeated measures. The model included the fixed effects of instrument, storage time, and their interaction and the random effect of steak. A Tukey-Kramer multiple comparison was performed to detect differences between means, with *p* ≤ 0.05 considered statistically significant. Data are presented as least-squares means ± standard error (SE). The degree of agreement between the two instruments was evaluated by a Bland–Altman limits of agreement analysis [17,18]. Instrument precision was evaluated by the coefficient of variation (CV). For the data obtained from the beef samples, the CV for each parameter was calculated at each time point and compared among instruments. Additionally, an overall CV value was calculated for *L**, *a**, and *b** collected on each of the three standard tiles.

## 3. Results and Discussion

### 3.1. Beef Color

Lean color strongly influences the consumer’s perception of beef quality, with bright, cherry-red beef being more desirable than dark-red beef [7]. This illustrates the importance of classifying fresh beef products based on lean color in order to satisfy consumers and assure repeat purchases. Indeed, the US Department of Agriculture beef quality grading system uses the color of the ribeye (*longissimus thoracis*; at the 12th rib) as an indicator of carcass maturity and palatability. Spectrophotometers and colorimeters are commonly used for the assessment of beef color by converting color attributes into quantifiable color coordinates (*L**, *a**, and *b**) defined by the CIE. In this three-dimensional color space, *L** refers to the lightness of the object (ranging from 0 for black to 100 for white), *a** indicates redness (+*a**) or greenness (−*a**), and *b** is a measure of yellowness (+*b**) or blueness (−*b**). From *a** and *b** values, *C**, an indicator of color intensity, and *h**, the angle at which a vector radiates into the red-yellow quadrant, can be computed.

In the current study, we evaluated the performance of the Color Muse Spectro 1 in assessing fresh beef color during storage as compared to the HunterLab MiniScan XE Plus spectrophotometer (Figure 2). Because these spectrophotometers can only evaluate a restricted area of the meat surface, multiple measurements are usually collected and averaged to represent the overall color of the meat. It is recommended to collect at least three-color scans (technical replicates) per sample, which are then averaged to give the mean value of *L**, *a**, and *b** [9]. However, Wei et al. [14] recommended that when Spectro 1 is used to evaluate meat color, four to eight technical replicates should be collected to ensure sufficient precision. Thus, we chose to collect seven technical replicates per steak in this study. An instrument × time interaction was detected for *L** (*p* = 0.0005, Figure 2A). Spectro 1 had greater *L** values at 0 and 3 days of storage than MiniScan (*p* ≤ 0.05), but no difference in *L** was observed between the two instruments at 5 days of storage. While no interaction effect was observed for *a** (Figure 2B), *b** (Figure 2C), *C** (Figure 2D), or *h** (Figure 2E), all of these parameters were significantly affected by the instrument (*p* ≤ 0.002) and time (*p* < 0.0001). Redness values collected with Spectro 1 at 0, 3, and 5 days of storage were all greater than those of MiniScan (*p* ≤ 0.01). The mean *b** value obtained with Spectro 1 at 3 days of storage was lower than the value of MiniScan (*p* = 0.006), but no differences in *b** values were detected between the two devices at 0 and 5 days of storage. At 0 day, Spectro 1 had a greater *C** value than MiniScan (*p* = 0.0004). However, there were no differences in *C** values at 3 and 5 days of storage. Conversely, lower *h** values were obtained with Spectro 1 throughout the storage period compared to MiniScan (*p* ≤ 0.004).

In comparison to MiniScan, Spectro 1 generally produced comparable *b** and *C** values, but greater *L** and *a** and lower *h** values. Consumers’ perception of fresh beef color is largely determined by the coordinates *L** and *a** [19]. For example, beef with low *L** and *a** values (i.e., dark cutter) is often rejected by the consumers due to its unappealing appearance [20]. The average *L** value obtained with Spectro 1 over the storage period was ~1 unit greater than that of MiniScan, which is considered a negligible difference [21]. On the other hand, a larger average difference in *a** (3.4 units) was observed between Spectro 1 and MiniScan. While absolute *a** values were different between the two instruments, Spectro 1 was capable of detecting the change in *a** values during storage. For instance, *a** decreased with increasing storage time (Figure 2B), an effect that was detected by both instruments. Differences in *a** between Spectro 1 and MiniScan could be partially attributed to the use of a different source of illumination in each instrument [22,23], with MiniScan utilizes a Xenon light source while Spectro 1 uses LED light. In addition, Spectro 1 utilizes a 45/0° measurement geometry, which may under- or overestimate color values depending on the texture of the surface. MiniScan, by contrast, operates based on a d/0 measurement geometry, which is less affected by surface texture [23]. Another factor that could have contributed to this disparity is the different calibration protocols used for each instrument.

Hue angle is another parameter that showed significant disagreement between Spectro 1 and MiniScan. The average *h** value obtained with Spectro 1 was 8.4 units lower than that of MiniScan. Given that *h** is computed from *a** and *b**, variation in *h** values is likely due to differences in *a** and *b** values among the two devices. In general, a high *a** value accompanied by a low *b** value generates a low *h** value [24,25]. In this study, lower numerical *b** values and greater *a** values were obtained with Spectro 1 throughout the storage period.

### 3.2. Percentages of Myoglobin Forms and MRA

Spectrophotometers measure the intensity of the reflected light at a wavelength ranging from 400 to 700 nm, while tristimulus colorimeters measure reflected light only at three wavelengths (red, green, and blue). Reflectance spectra obtained with spectrophotometers allow for the determination of the relative proportions of myoglobin redox forms at the meat surface along with color coordinates, providing more depth to the evaluation of meat color [26]. In addition, MRA, which indicates the inherent ability of meat to convert MMb back to DMb, can also be evaluated utilizing the spectral analysis capability of spectrophotometers. In contrast, colorimeters are only capable of providing color coordinates. In this study, percentages of myoglobin forms and MRA were compared between Spectro 1 and MiniScan throughout the 5-days storage period. No interaction or instrument effect was seen for MMb% (Figure 3A). Yet, a significant time effect was found (*p* < 0.0001), with MMb% gradually increasing throughout the storage period. Deoxymyoglobin% was significantly affected by the interaction between instrument and time (*p* = 0.002, Figure 3B). Deoxymyoglobin% values obtained with Spectro 1 at 0 and 3 days of storage were lower than their MiniScan counterparts (*p* ≤ 0.003), but similar values were obtained at 5 days. A significant time effect was observed for OMb% (*p* < 0.0001, Figure 3C); OMb% values decreased with increasing the storage period. There was an instrument × time interaction with respect to MRA% (*p* = 0.002; Figure 4). There were no differences in MRA% between the two devices at 0 and 5 days, yet, Spectro 1 had a greater value at 3 days.

The oxidized form of myoglobin, MMb, is often associated with product acceptability. Previous research shows that most consumers reject retail meats when surface browning, due to MMb accumulation, exceeds 40% [2]. Therefore, accurately estimating the fraction of myoglobin present as MMb in the meat during storage and display is essential for mitigating financial losses. The MRA is another property that is crucial for meat color stability during storage and display. Our data indicate that MMb%, OMb%, and MRA% values obtained with Spectro 1 throughout the storage period were comparable to MiniScan. Furthermore, Spectro 1 was capable of detecting changes in the percentages of myoglobin forms and MRA% during storage. In this study, OMb% and MRA% decreased with increasing storage time, whereas an increase in MMb% was seen, which agrees with what is typically known about the changes in myoglobin redox forms [27] and MRA% [28] during beef storage. However, Spectro 1 had generally lower DMb% values than the MiniScan. This finding was not surprising as *a** is typically inversely related to DMb% [29], an effect that was observed in this study. The lack of agreement concerning DMb% between Spectro 1 and MiniScan could be due to the inherent differences between the two instruments that we discussed in the previous section. There is also the possibility that the different calibration procedures used in each instrument could have contributed to this difference.

### 3.3. Agreement between Instruments

Bland–Altman plots were used to examine the agreement between Spectro 1 and MiniScan for color coordinates (Figure 5). In a Bland–Altman scatter plot, the x-axis represents the mean of the measurements, while the y-axis is the difference between two paired measurements. This graphical tool provides the mean of the differences between two measurement devices/methods (bias) and the lower and upper limits of agreement at a 95% confidence level (mean difference ± 1.96 × standard deviation (SD)). However, it is important to note that the plot itself does not establish agreement but quantifies the bias and ranges of agreement within 95% confidence intervals. Acceptable limits should be determined by the research group, typically, with reference to previous literature [17]. If a sufficient agreement range is observed, then the instruments/methods may be considered interchangeable for that measurement.

The bias for *L** was 0.84, with 95% limits of agreement ranging from −3.14 to 4.82 (Figure 5A). In other words, Spectro 1 measured on average 0.84 units greater than MiniScan, with 95% of the differences within −3.14 to 4.82 *L** units. The small bias coupled with relatively narrow limits of agreement suggest that *L** can be interchangeable between the two instruments. The Bland–Altman plots for *b** (Figure 5C) and *C** (Figure 5D) also showed small bias and narrow limits of agreement. In contrast, relatively large bias and wider limits of agreement were observed for *a** (Figure 5B) and *h** (Figure 5E). The limits of agreement for *a** and *h** did not include zero, indicating that Spectro 1 consistently measured greater *a** and lower *h** values in comparison to MiniScan. Curiously, there appears to be an increase in the difference between the two instruments for *a** as mean values increase (x-axis). As such, caution should be practiced when comparing *a** and *h** between these two instruments. In totality, the Bland–Altman analysis demonstrated good agreement between Spectro 1 and MiniScan for color coordinates.

Bland–Altman limits of agreement analysis was also performed for myoglobin redox forms and MRA% (Figure 6). Although the bias was small for MMb% (Figure 6A), OMb% (Figure 6C), and MRA% (Figure 6D), the ranges of agreement were wide for all myoglobin redox forms and MRA%, which suggests that these measurements exhibited high variability among the two instruments. The average mean difference for MMb% (3.52) indicates that Spectro 1 had overall greater MMb% values; however, a reduction in the differences was observed with increasing MMb% mean values. On the other hand, Spectro 1 generally measured less OMb% than MiniScan when the mean values were below ~25 or above ~75, but measured greater values between 25 and 75. No specific pattern was observed for DMb% (Figure 6B) and MRA%. Overall, myoglobin redox forms and MRA% showed variability among the two instruments. This does not suggest that Spectro 1 cannot be used to evaluate myoglobin redox forms and MRA in beef. Rather, it indicates that these parameters are not interchangeable between Spectro1 and MiniScan.

### 3.4. Precision and Stability

Precision is defined as the agreement between replicate measurements of the same sample, which can be evaluated by calculating the CV. As a general rule, a CV value of <5% indicates low variation [30]. To assess the precision of Spectro 1 in evaluating beef color, we compared the CV values of all color coordinates, myoglobin redox forms, and MRA% between the two instruments (Table 1). The CV of each parameter was calculated from the technical replicates of each steak and then averaged across the 12 steaks. Comparable CV values were seen between the two devices for all color coordinates at each time point, except for *h** at 5 days of storage, in which Spectro 1 had significantly greater CV than MiniScan (*p* = 0.008). It is noteworthy that a general increase in the CV was observed for all color coordinates as storage time increased. Using the HunterLab MiniScan XE Plus, King et al. [28] reported similar CV values for beef color coordinates to those obtained in this work. There were no differences in the CV between Spectro 1 and MiniScan for MMb% at 3 and 5 days, DMb% at 0 and 5 days, and OMb% at all time points. However, Spectro 1 had a lower CV for MMb% at 0 day (*p* = 0.002) and a greater CV for DMb% at 3 days of storage (*p* = 0.0004). No differences were detected in the CV values for MRA% at 0 and 5 days, though, a significantly lower CV for MRA% was observed for Spectro 1 at 3 days (*p* = 0.001).

Overall, the CV values obtained in this study are considered high, indicating variations between measurements. Meanwhile, differences in the CV between the two instruments were minimal. Considering the fact that beef is heterogeneous in color, it is likely that the high CV values obtained in this study reflect differences in color within and among steaks. Variation in measurements may also result from data drifting (i.e., instrument instability) over time [31]. In the present study, the stability of the instruments was evaluated by comparing the Δ*E***_ab_* of three standard tiles over a period of 60 min (Figure 7). Our data indicate that both instruments maintained stable color coordinates over the 60-min period for the white (Figure 7A), blue (Figure 7B), and green (Figure 7C) tiles, with Δ*E***_ab_* ranging between 0.02 and 0.23 for Spectro 1 and 0.07 and 0.21 for MiniScan. Kirillova et al. [32] indicated that a Δ*E***_ab_* value of <3 is barely perceived visually, suggesting that data fluctuation was negligible for Spectro 1 and MiniScan. The same data used for Δ*E***_ab_* calculation were also used to compute an overall CV for *L**, *a**, and *b** of each tile (Table 2). The CV for *L** was not different between the two devices for the white tile, but Spectro 1 had a greater *L** CV for the blue (*p* = 0.003) and green tiles (*p* = 0.04). On the other hand, Spectro 1 had lower *a** and *b** CV values for all tiles (*p* < 0.0001). Although there were differences in CV between the two instruments, both showed small CV values for all color coordinates for each of the three tiles, indicating high precision. These results confirm that the large variation in the beef color components were likely due to color variation across the surface of the steaks.

## 4. Conclusions

In this research, we demonstrated that Spectro 1 generates similar *L**, *b**, *C**, MMb%, OMB%, and MRA% values to those of MiniScan. Yet, *a** values measured with Spectro 1 were overestimates compared to MiniScan, whereas *h** and DMb% were underestimates. Regardless, Spectro 1 was capable of detecting changes in color coordinates, myoglobin redox forms, and MRA% throughout the storage period. Bland–Altman analysis indicated that *L**, *b** and *C** values are interchangeable between Spectro 1 and MiniScan, whereas *a**, *h**, myoglobin redox forms, and MRA% are not. Thus, caution should be practiced when comparing color data between the two instruments. Moreover, Spectro 1 showed excellent stability in measuring color coordinates over time, as evidenced by the low Δ*E***_ab_*. In totality, Spectro 1 is an affordable spectrophotometer that has the potential to be used in beef color research.

## Figures and Tables

**Figure 1 foods-11-02091-f001:**
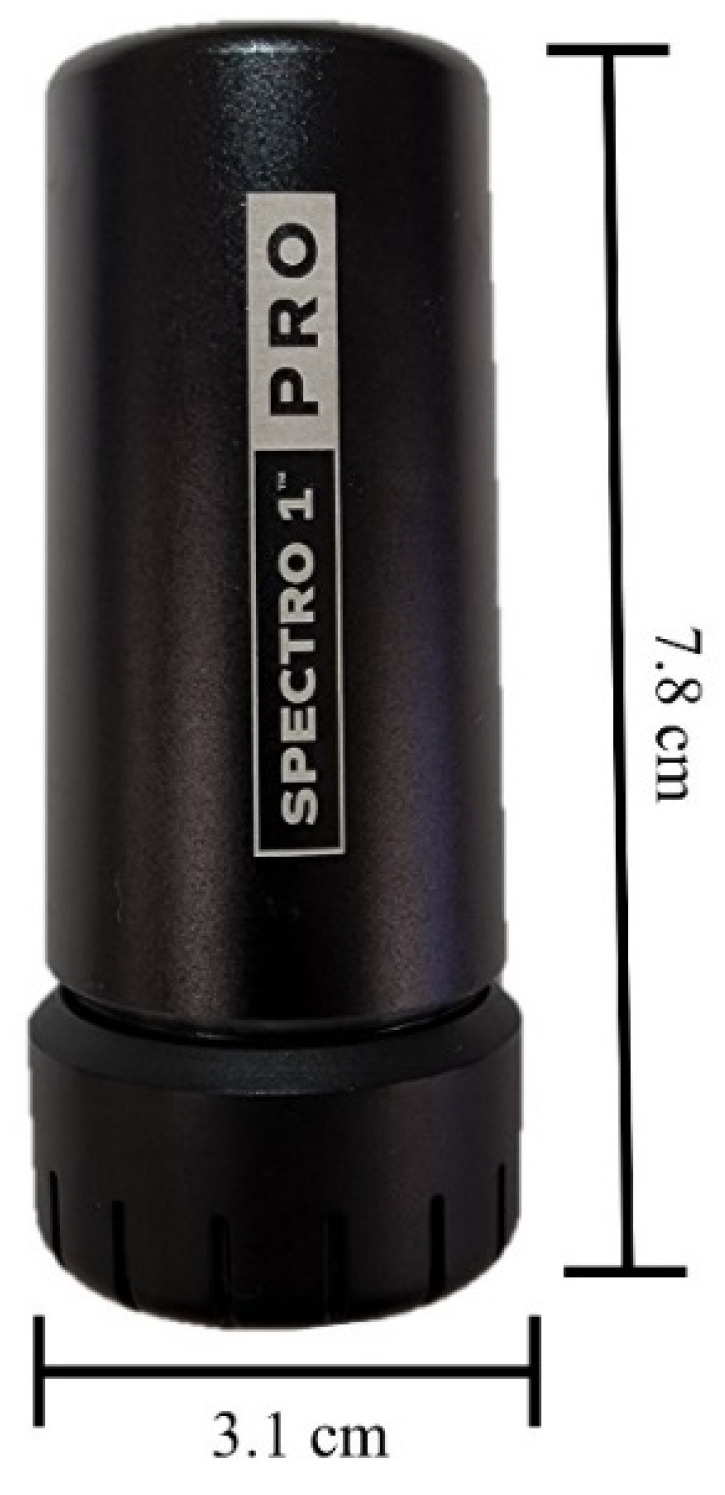
Color Muse Spectro 1 along with its dimensions.

**Figure 2 foods-11-02091-f002:**
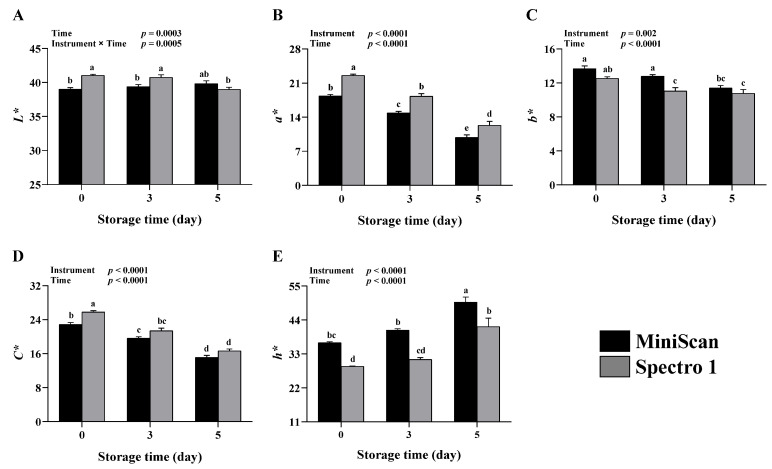
Lightness (*L**; (**A**)), redness (*a**; (**B**)), yellowness (*b**; (**C**)), chroma (*C**; (**D**)), and hue (*h**; (**E**)) values obtained from beef steaks at 0, 3, and 5 days of storage using Spectro 1 and MiniScan (*n* = 12). Data are least-squares mean ± SE. ^a–e^ Means lacking a common letter differ significantly (*p* ≤ 0.05).

**Figure 3 foods-11-02091-f003:**
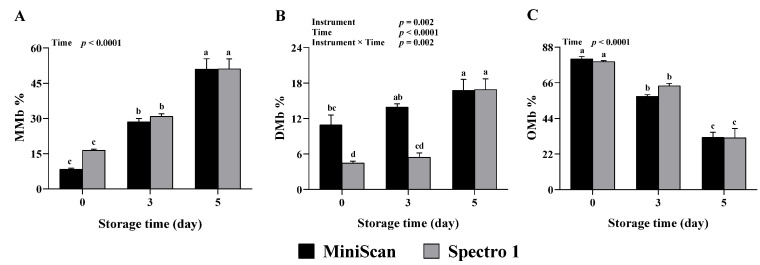
Surface metmyoglobin% (MMb%; (**A**)), deoxymyoglobin% (DMb%; (**B**)), and oxymyoglobin% (OMb%; (**C**)) values obtained from beef steaks at 0, 3, and 5 days of storage using Spectro 1 and MiniScan (*n* = 12). Data are least-squares mean ± SE. ^a–d^ Means lacking a common letter differ significantly (*p* ≤ 0.05).

**Figure 4 foods-11-02091-f004:**
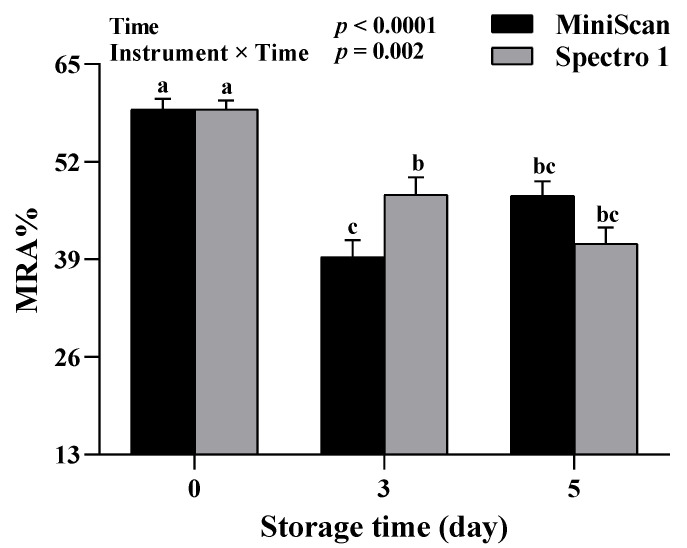
Metmyoglobin reducing ability% (MRA%) values obtained from beef steaks at 0, 3, and 5 days of storage using Spectro 1 and MiniScan (*n* = 12). Data are least-squares mean ± SE. ^a–c^ Means lacking a common letter differ significantly (*p* ≤ 0.05).

**Figure 5 foods-11-02091-f005:**
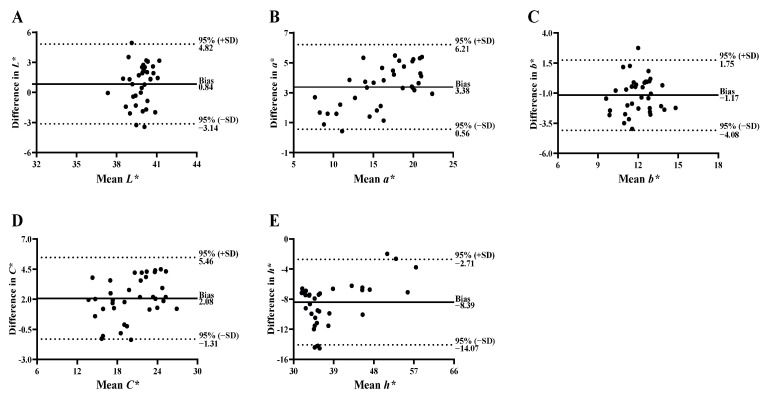
Bland-Altman limits of agreement of lightness (*L**; (**A**)), redness (*a**; (**B**)), yellowness (*b**; (**C**)), chroma (*C**; (**D**)), and hue (*h**; (**E**)) values obtained from beef steaks using Spectro 1 and MiniScan.

**Figure 6 foods-11-02091-f006:**
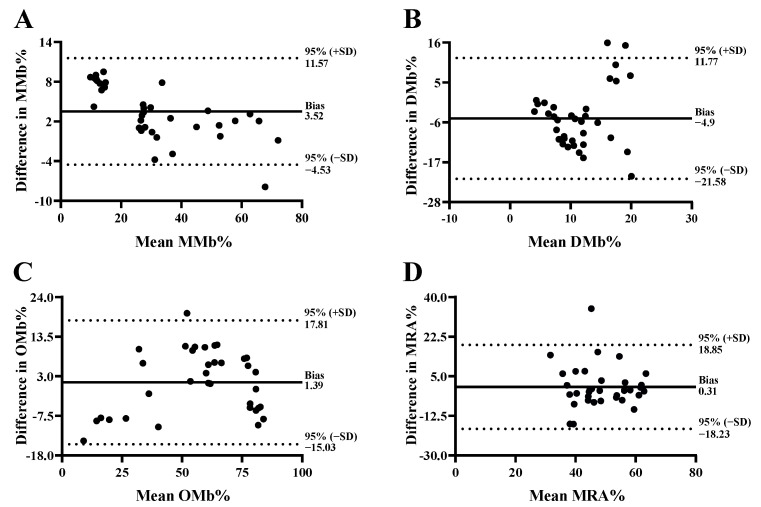
Bland-Altman limits of agreement of metmyoglobin% (MMb%; (**A**)), deoxymyoglobin% (DMb%; (**B**)), oxymyoglobin% (OMb%; (**C**)), and metmyoglobin reducing ability% (MRA%; (**D**)) values obtained from beef steaks using Spectro 1 and MiniScan.

**Figure 7 foods-11-02091-f007:**
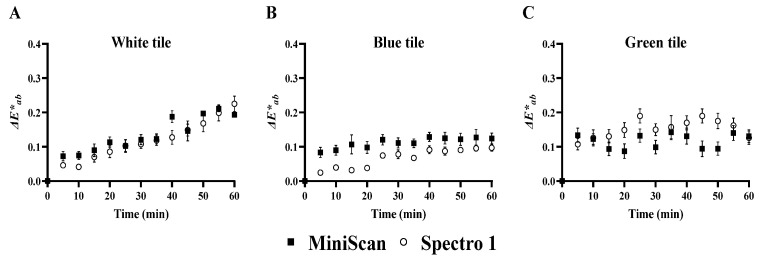
Scatter plots of total color difference over time (Δ*E***_ab_*) for Spectro 1 and MiniScan. Measurements were collected on three different calibration tiles: white tile (**A**), blue tile (**B**), and green tile (**C**). Values are expressed as the difference in color values at each time point to measurement taken at 0 min.

**Table 1 foods-11-02091-t001:** Coefficient of variation (CV) for lightness (*L**), redness (*a**), yellowness (*b**), chroma (*C**), hue (*h**), metmyoglobin% (MMb%), deoxymyoglobin% (DMb%), oxymyoglobin% (OMb%), and metmyoglobin reducing ability% (MRA%) obtained from beef steaks at 0, 3, and 5 days of storage using Spectro 1 and MiniScan (*n* = 12). Data are least-squares mean ± SE.

Storage Time	Parameter								
	*L**	*a**	*b**	*C**	*h**	MMb%	DMb%	OMb%	MRA%
0 day									
Spectro 1	3.57	6.77	11.63	7.84	4.63	9.75	55.71	2.77	5.06
MiniScan	3.78	8.68	10.43	8.88	3.56	26.23	55.35	5.95	12.19
SE	0.27	0.72	0.88	0.75	0.45	2.45	2.90	0.53	1.81
*p*-value	0.99	0.90	0.99	0.99	0.98	0.002	0.99	0.94	0.95
3 days									
Spectro 1	4.58	9.89	10.52	8.64	7.60	14.11	61.97	8.34	5.62
MiniScan	4.51	11.91	10.97	11.23	3.35	16.38	25.61	9.55	41.24
SE	0.39	0.94	0.82	0.79	0.73	1.26	4.72	0.84	7.55
*p*-value	0.99	0.87	0.99	0.61	0.09	0.99	0.0004	0.99	0.001
5 days									
Spectro 1	4.70	16.01	13.38	11.66	12.14	14.11	40.94	33.97	4.29
MiniScan	4.70	16.51	11.45	12.76	6.43	22.00	45.21	30.19	6.63
SE	0.38	1.02	1.34	0.90	1.26	2.65	5.17	2.62	0.72
*p*-value	0.99	0.99	0.94	0.98	0.008	0.35	0.99	0.87	0.99

**Table 2 foods-11-02091-t002:** Coefficient of variation (CV) for lightness (*L**), redness (*a**), and yellowness (*b**) obtained from white, blue, and green calibration tiles over a 60-min period using Spectro 1 and MiniScan (*n* = 12). Data are least-squares mean ± SE.

Standard Tile	Color Coordinate		
	*L**	*a**	*b**
White			
Spectro 1	0.041	1.728	0.373
MiniScan	0.035	3.560	1.332
SE	0.007	0.257	0.109
*p*-value	0.6448	<0.0001	<0.0001
Blue			
Spectro 1	0.026	0.102	0.176
MiniScan	0.017	0.299	0.309
SE	0.001	0.026	0.018
*p*-value	0.003	<0.0001	<0.0001
Green			
Spectro 1	0.048	0.178	0.247
MiniScan	0.038	0.383	0.567
SE	0.003	0.023	0.041
*p*-value	0.04	<0.0001	<0.0001

## Data Availability

The data presented in this study are available on request from the corresponding author.

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
