# Peer review of "Spectro 1—A Potential Spectrophotometer for Measuring Color and Myoglobin Forms in Beef"

_foods, 2022, doi:10.3390/foods11142091_

Round 1

Reviewer 1 Report

Abstract

Line 17: 5-d post mortem storage period (use this term in the whole MS)

Line 21-22: the post mortem storage period

Line 24:  Color coordinates measured by Spectro 1 exhibited excellent stability over time- what do the authors want to mean by this part? The Spectro 1 is perfect one to measure color coordinates or the meat color is stable?

Line 26: Potential for beef color only but not for myoglobin redox, right? please make it clear. I suggest to use meat color rather than beef color as conclusion.

Introduction:

Line 34: delete fresh

Line 63: Mention which application of smartphone?

Line 66: What are the other devices (mention few of them) and what are the capabilities (mention here).

Line 72-73: the objective of this study was to investigate the efficacy of the Color Muse Spectro 1 in measuring fresh beef surface color and myoglobin redox forms. Do the authors considered 5-d post mortem aged beef as fresh???

Material and Methods:

This is the main part of this MS because of the validation procedure of the instrument as well as the methods followed for beef color measurement.

Line 79: Experimental design and materials

Line 80-87:

Do the authors know when the animals were killed, how long the carcasses were kept hanging in slaughterhouse to resolve the rigor before the carcass fabrication, how long the steaks were displayed at retail before purchasing for this study, how the steaks were displayed during the retail display, are all the steaks from same animal, how the steaks were chosen for each group??

Why the steak was overwrapped with oxygen permeable PVC film for storage? Is it to follow the retail display? The authors should specify these things in their methods section.

How the authors determine 0, 3 and 5 d of postmortem storage?

Line 94:

Is this the way researchers of meat science do blooming of meat? Without fresh cut of the meat pieces , bloom will not be done. According to this line, packages open only for blooming. The meat steaks were packaged with permeable PVC which is already bloomed and developed met-myoglobin for 3 and 5 d storage steaks on the surface of the steaks and the reviewer think this is not correct way to do.

Line 98: Is this instrument requires to calculate chroma and hue values separately from L*, a* and b*? There are many instruments for color measurement which could also measure C* and hue along with L*, a* and b* values.

Line 106: It is metmyoglobin reducing activity but then in line 107 you started with MRA which indicates myoglobin reducing activity- Which one is correct?

Line 107: Don't start a sentence with abbreviation.

Line 110, 113: Scan and rescan at which wavelengths.

Line 114: The equation shows  Initial MMB is greater than Final MMB, is that correct?

Line 163, 167, 168, 170: 0 and 3 d of storage

Line 169, 171: b* values, C* values

Line 215: 5-d, delete - from here to follow same style.

Line 217, 218: Don't start a sentence with abbreviation.

Figure 4: MRA% at 0 d storage period, seems between two instruments samples showed same activity. Is this the mean of samples you considered? Please use n=.... to show number of samples used for each figure.

Line 423: Delete A after number 9.

Reviewer 2 Report

The review of the manuscript  (ID: foods- 1798374) entitled: “Spectro 1 – A Potential Spectrophotometer for Measuring Color and Myoglobin Forms in Beef”.

Dear Authors,

with great interest I read the article that was aimed at to compare the Color Muse Spectro 1 sensor to the HunterLab MiniScan XE Plus spectrophotometer for evaluating beef color.

Color is one of the most important attributes of meat that is taken into consideration by consumers during retail purchases. There is indeed a need to improve color measurement methods, particularly in industrial applications. Currently, the most commonly used instruments for color measurements are the Minolta (Chiyoda City, Tokyo, Japan) and the HunterLab (Reston, VA, USA). Most of the researchers (over 60%) who publish their results use Minolta for the measurements to meat color, and the NPPC (Official Color and Marbling Standards) assigns Minolta L * values to color standards, rather than Hunter Lab L * values. So please explain why your research used the HunterLab MiniScan XE Plus spectrophotometer.

In addition, I ask the authors to describe in more detail, both for the purpose and in conclusions, the practical implications of the possible use of the Color Muse Spectro 1 sensor. Is the long calibration time required by the Spectro 1 series instruments after many measurements a real obstacle to its widespread use?
